# Estimation and Clustering in Finite Mixture Models: Bayesian Optimization as an Alternative to EM

## Abstract

We address the problem of maximum likelihood estimation (MLE) for finite mixtures of elliptically distributed components, a setting that extends beyond the classical Gaussian mixture model. Standard approaches such as the Expectation–Maximization (EM) algorithm are widely used in practice but are known to suffer from local optima and typically require strong assumptions (e.g., Gaussianity) to guarantee convergence. In this work, we use the Bayesian Optimization (BO) framework for computing the MLE of general elliptical mixture models. We establish that the estimates obtained via BO converge to the true MLE, providing asymptotic *global* convergence guarantees, in contrast to EM. Furthermore, we show that, when the MLE is consistent, the clustering error rate achieved by BO converges to the optimal misclassification rate. Our results demonstrate that BO offers a practical, flexible, and theoretically sound alternative to EM for likelihood-based inference in mixture models, particularly in complex and/or non-Gaussian elliptical families where EM is difficult to implement and/or analyze. Experiments on synthetic and real data sets confirm the effectiveness and practical applicability of BO as an alternative to EM.

## 1 Introduction

Finite mixture models are a fundamental ingredient in statistical modeling for applications such as clustering, density estimation, and anomaly detection (McLachlan et al., 2019). A central task in this context is the maximum likelihood estimation (MLE) of the mixture parameters, which provides a principled and statistically efficient route to inference in general models.

Gaussian mixture model (GMM) is the class of mixture models that has received most of the attention so far. However, many real datasets exhibit heavy tail distributions, skewness and/or robustness requirements that cannot be adequately captured by a Gaussian model. Mixtures of elliptical distributions, such as Student's t or more general families, provide a natural extension that models better such constrains. However, estimating such mixtures is a longstanding challenge: the likelihood surface is highly non-convex, and standard algorithms are prone to local optima. Maximizing the likelihood of a finite mixture model is generally intractable because of the presence of latent component assignments.

Expectation-Maximization (EM) algorithm is currently the standard approach for maximum likelihood estimation in finite mixture models. It iteratively alternates between computing posterior responsibilities for each component (E-step) and updating the parameters to maximize the expected complete-data log-likelihood (M-step). The EM algorithm comes with two important advantages for Gaussian mixtures: it is computationally efficient and it admits closed-form updates. Unfortunately, these advantages do not extend in general to non-Gaussian distributions, such as the Student's t or skewed families. In particular, the presence of heavy tails or skewness makes the M-step much more complex, often requiring the introduction of auxiliary variables to preserve tractable updates. For Student's t mixture models (SMM), Peel & McLachlan (2000) address this issue by introducing a latent scale variable for each observation, which allows the complete-data log-likelihood to resemble that of a Gaussian mixture with scaled covariances. This formulation leads to a robust variant of the EM algorithm that improves estimation under outliers or heavy-tailed data. Similar latent-variable

representations and modified EM schemes have been proposed for skewed distributions, but they often involve additional approximations or numerical steps, increasing the complexity of the algorithm and making it more sensitive to initialization (Lin, 2010; Lee & McLachlan, 2016). These issues with EM motivate the exploration of alternative optimization strategies for mixture models that are more complex than GMM.

In this work, we propose Bayesian Optimization (BO) as a principled alternative for computing the MLE in mixture models. BO is a global optimization framework designed for black-box objectives, combining surrogate modeling with adaptive exploration–exploitation strategies. These features make BO particularly well-suited to the likelihood maximization problem, where gradients may be unreliable or even inaccessible, and where the objective landscape is highly multi-modal. Importantly, and contrary to EM, BO is designed to escape local traps and to adaptively refine the search for the global maximum. However, optimizing likelihoods of general elliptical mixtures using BO remains challenging in practice: the parameter space includes positive semidefinite shape matrices, and the likelihood is invariant under permutation of the mixture components, resulting in multiple equivalent global optima.

We overcome these difficulties and we establish rigorous convergence guarantees for BO in the context of elliptical mixture models. Specifically, we show that the sequence of estimates produced by BO converges to the MLE. Moreover, when the MLE is consistent, we prove that the clustering risk of the BO-based estimator converges to the asymptotic optimal misclassification rate. These are the first asymptotic *global* convergence guarantees of a practically implementable algorithm for mixtures of general elliptical families.

We complement these theoretical results with experiments on data generated from mixtures of Student's t distributions. We find that BO consistently outperforms standard clustering algorithms such as k-means, spectral clustering, and EM, which confirms the practical benefits of BO with highly non-convex likelihood landscapes. We next make experiments on real-world datasets, which highlights the flexibility and robustness of BO in applied settings. Overall, our work establishes BO as a practical and theoretically grounded tool for maximum likelihood estimation in complex mixture models.

The paper is structured as follows. Section 2 reviews background material on mixture models and BO. Section 3 presents our approach for using BO to compute the MLE and establishes the corresponding theoretical guarantees. Section 4 reports the results of our numerical experiments. Finally, Section 5 concludes the paper and outlines directions for future work.

**Notations** We denote by $\mathcal{S}_k$ the group of permutations over $[k] = \{1, \cdots, k\}$. For a vector $u \in \mathbb{R}^k$ and a permutation $\sigma \in \mathcal{S}_k$, we let $\sigma \circ u = (u_{\sigma(1)}, \cdots, u_{\sigma(k)})$, that is, the vector $u$ with its entries permuted according to $\sigma$. We let $\mathcal{S}_{++}^d$ denote the cone of $d \times d$ positive definite matrices, and $\Delta^{k-1}$ the probability simplex, i.e., $\Delta^{k-1} = \{\pi \in [0,1]^k \colon \sum_{a=1}^k \pi_a = 1\}$.

## 2 BACKGROUND

### 2.1 FINITE MIXTURE MODELS

We consider a parametric family $\mathcal{F} = \{f(x; \theta), \theta \in \Theta\}$ of probability distributions over $\mathbb{R}^d$. We suppose that $\mathcal{X} \subseteq \mathbb{R}^d$ and that the parameter space $\Theta$ is equipped with a metric $\rho$. A *finite mixture model* with $k \geq 1$ components is defined by the probability distribution $M$ such that

$$M(x; \pi, \theta) = \sum_{a=1}^k \pi_a f(x; \theta_a), \tag{1}$$

where $\pi \in \Delta^{k-1}$ is the vector of the mixing proportions and $\theta \in \Theta^k$ are the parameters associated to the components of the mixture. We will always assume that (i) the mixture family is identifiable, that is, if $M(x; \pi, \theta) = M(x; \pi', \theta')$ for almost all $x$, then there exists a permutation $\sigma \in \mathcal{S}_k$ such that $\pi' = \sigma \circ \pi$ and $\theta' = \sigma \circ \theta$, and (ii) that the mixture has exactly $k$ components, that is $\min_{a \in [k]} \pi_a > 0$.

A common choice for the parametric family $\mathcal{F}$ is the set of multivariate Gaussian distributions, leading to the widely used Gaussian Mixture Model (GMM). However, we also consider more general

families of distributions for the component densities, such as the multivariate Student's t-distribution, which introduces greater robustness and flexibility. Tables 1 and 2 in Appendix B summarize some common non-skewed and skewed parametric distributions, and we refer to Azzalini & Capitanio (2003); Sahu et al. (2003); Lin (2010) for additional details on skewed elliptic distributions and their applications.

**Parameter Estimation.** A long line of recent work has investigated the convergence rates of finite mixture models under varying degrees of identifiability of the true mixing distribution $M^*$ (Nguyen, 2013; Ho & Nguyen, 2016a;b; Heinrich & Kahn, 2018). These studies typically focus on the Maximum Likelihood Estimator (MLE), which, given an i.i.d. sample $X_1, \cdots, X_n$ from $M^*$, is given by $\hat{M}^{\mathrm{MLE}}(x) = \sum_{a=1}^k \hat{\pi}_a^{\mathrm{MLE}} f(x; \hat{\theta}_a^{\mathrm{MLE}})$ where

$$(\hat{\pi}^{\mathrm{MLE}}, \hat{\theta}^{\mathrm{MLE}}) = \underset{\substack{\theta \in \tilde{\Theta} \\ \pi \in \Delta^{k-1}}}{\arg\max} \sum_{i=1}^n \log \left( \sum_{a=1}^k \pi_a f(X_i; \theta_a) \right), \tag{2}$$

where $\tilde{\Theta}$ is a compact subset of $\Theta^k$ that contains $\theta^*$ and on which the log-likelihood is bounded. In particular, Ho & Nguyen (2016b) show that, under technical assumptions on the parametric family $\mathcal{F}$, we have $\mathbb{E}\left[W_1(M^*, \hat{M}_n^{\mathrm{MLE}})\right] \leq (\log n/n)^{1/2}$ and $\mathbb{E}\left[W_2(M^*, \hat{M}_n^{\mathrm{MLE}})\right] \leq (\log n/n)^{1/4}$, where $W_r$ is the Wasserstein distance of order $r$.

The EM algorithm is widely used for parameter maximum likelihood estimation in Gaussian mixture models. While extensions of EM have been proposed for more complex families (such as Student or skewed-Student mixtures, as noted in the introduction), rigorous theoretical guarantees remain largely restricted to simpler cases. In fact, even for mixtures of three isotropic, well-separated Gaussians, Jin et al. (2016) showed that the likelihood surface contains arbitrarily bad local maxima, implying that EM with random initialization is likely to get trapped in suboptimal solutions. Nevertheless, some theoretical guarantees have been established in special settings. For instance, in mixtures of two isotropic Gaussians, Wu & Zhou (2021) proved that EM with random initialization converges within $\mathcal{O}(\sqrt{n})$ iterations with high probability and achieves parameter estimates at the minimax rate. Moreover, under suitable separation conditions on the components, Zhao et al. (2020) demonstrated that EM, when initialized sufficiently close to the true centers, converges linearly to the global optimum of the log-likelihood.

**Bayes Optimal Clustering** An i.i.d. sample $X_1, \cdots, X_n$ from a mixture $M^*(\cdot; \pi^*, \theta^*)$ can be augmented with latent variable $z_1^*, \cdots, z_n^*$ such that $z_1^*, \cdots, z_n^*$ is an i.i.d. sample from $\mathrm{Multi}([k], \pi^*)$ and $X_i \mid z_i^* \sim f(\cdot \mid \theta_{z_i}^*)$. The task of recovering the latent variables $z_1^*, \cdots, z_n^*$ given the observation of $X_1, \cdots, X_n$ is called *clustering*. The misclustering error of an estimator $\hat{z}$ of $z^*$ is defined by the fraction of disagreements between $\hat{z}$ and $z^*$, up to a global permutation of the labels of $\hat{z}$, i.e.,

$$\mathrm{loss}(z^*, \hat{z}) = \frac{1}{n} \min_{\sigma \in \mathcal{S}_k} \sum_{i \in [n]} \mathbb{1}\{z_i^* \neq \sigma(\hat{z}_i)\}, \tag{3}$$

where $\mathcal{S}_k$ denotes the group of permutations on $[k]$. When the mixture parameters $\theta_1^*, \cdots, \theta_k^*$ are known, Dreveton et al. (2024) showed that the expected misclustering error of the best estimator is asymptotically of the order $\exp\left(-\min_{a \neq b \in [k]} \mathrm{Chernoff}(\theta_a^*, \theta_b^*)\right)$, where $\mathrm{Chernoff}(\theta_a^*, \theta_b^*)$ denotes the Chernoff information between the probability densities $f(\cdot; \theta_a^*)$ and $f(\cdot; \theta_b^*)$, given by

$$\mathrm{Chernoff}(\theta_a^*, \theta_b^*) = -\log \left( \inf_{s \in (0,1)} \int f^s(x; \theta_a^*) f^{1-s}(x; \theta_b^*) dx \right). \tag{4}$$

When the mixture parameters are unknown (which is typically the case in practice), Lu & Zhou (2016) show that the standard Lloyd's algorithm achieves this exponential rate if $M^*$ is a mixture of isotropic Gaussian distributions. More recently, Chen & Zhang (2024) and Dreveton et al. (2024) demonstrate that a modified Lloyd's algorithm attains the same rate in a mixture of anisotropic Gaussian distributions and in a mixture of Laplace distributions, respectively.

## 2.2 BAYESIAN OPTIMIZATION

BO is a framework that has proven to be successful at optimizing a costly-to-evaluate black-box function $f$ in a broad and diverse range of applications (Marchant & Ramos, 2012; Wang et al., 2014; Bardou et al., 2025). Using (a) a Gaussian Process (GP) as a surrogate model for the black-box function $f$, a BO algorithm exploits (b) an acquisition function to optimize $f$ in an online fashion. This policy is (c) guaranteed to globally maximize $f$ in the long run under mild assumptions. In this section, we discuss (a), (b), and (c) in detail.

**(a) Surrogate Model.** The goal of a BO algorithm is to maximize a black-box objective function $f : \tilde{\Theta} \subset \mathbb{R}^D \to \mathbb{R}$, where $\tilde{\Theta}$ is a compact search space. To do so, it leverages a stochastic process, which is in general a GP (Williams & Rasmussen, 2006), as a surrogate model for $f$. Formally, it operates under the assumption that $f \sim \mathcal{GP}(\mu, \kappa)$, where $\mu : \tilde{\Theta} \to \mathbb{R}$ and $\kappa : \tilde{\Theta} \times \tilde{\Theta} \to \mathbb{R}$ are the prior mean and covariance of the GP, respectively, such that for any $\theta, \theta' \in \tilde{\Theta}$, $\mu(\theta) = \mathbb{E}\left[f(\theta)\right]$ and $\kappa(\theta, \theta') = \mathrm{Cov}\left[f(\theta), f(\theta')\right]$. Most definitions of covariance functions $\kappa$ include hyperparameters that are inferred with MLE from the observations in $\mathcal{D}_t$. Without loss of generality, we set that for any $\theta \in \tilde{\Theta}$, $\mu(\theta) = 0$ and $\kappa(\theta, \theta) = 1$. Given a dataset $\mathcal{D}_t = \{(\theta_i, y_i)\}_{i \in [t]}$ of $t$ observations, where $y_i = f(\theta_i)$, $f|\mathcal{D}_t$ is also a GP. In particular, for any $\theta \in \tilde{\Theta}$, $f(\theta)|\mathcal{D}_t \sim \mathcal{N}\left(\mu_t(\theta), \sigma_t^2(\theta)\right)$ where

$$\mu_t(\theta) = \kappa(\theta, \mathcal{D}_t)^\top \kappa(\mathcal{D}_t, \mathcal{D}_t)^{-1} y, \tag{5}$$

$$\sigma_t^2(\theta) = \kappa(\theta, \theta) - \kappa(\theta, \mathcal{D}_t)^\top \kappa(\mathcal{D}_t, \mathcal{D}_t)^{-1} \kappa(\theta, \mathcal{D}_t), \tag{6}$$

where $\kappa(\mathcal{X}, \mathcal{X}') = (\kappa(\theta_i, \theta_j))_{\theta_i \in \mathcal{X}, \theta_j \in \mathcal{X}'}$ and $y = (y_1, \cdots, y_t)$.

Note that assuming $f \sim \mathcal{GP}(0, \kappa)$ is mild because GPs enjoy a universal approximation property. As an example, the posterior mean (5) of a GP equipped with the Gaussian (RBF) kernel can approximate any continuous function given a sufficiently large dataset $\mathcal{D}_t$ (Micchelli et al., 2006).

**(b) Acquisition Function.** At iteration $t + 1$, a BO algorithm must acquire a new observation $(\theta_{t+1}, y_{t+1})$ that improves the quality of the surrogate model (exploration) and such that $y_{t+1}$ is close to $\max_{\theta \in \tilde{\Theta}} \mu_t(\theta)$ (exploitation). To find a trade-off between these two objectives, a BO algorithm uses an acquisition function $\varphi_t : \tilde{\Theta} \to \mathbb{R}$ and sets $\theta_{t+1} = \arg\max_{\theta \in \tilde{\Theta}} \varphi_t(\theta)$. There are many popular acquisition functions, such as GP-UCB (Srinivas et al., 2012), Expected Improvement (Mockus, 1994) or Knowledge Gradient (Gupta & Miescke, 1996).

**(c) Asymptotic Performance.** The optimization error of a BO algorithm at iteration $t$ is measured by the instantaneous regret $r_t = f(\theta^*) - f(\theta_t) \geq 0$, where $\theta^* = \arg\max_{\theta \in \tilde{\Theta}} f(\theta)$. The cumulative regret $R_T = \sum_{t=1}^{T} r_t$ quantifies the optimization error from the beginning of the optimization process and up to iteration $T$. A BO algorithm is said to have the no-regret property if it satisfies $\lim_{T \to \infty} R_T/T = 0$, that is, $R_T \in o(T)$. A no-regret BO algorithm is guaranteed to globally maximize its objective function $f$ asymptotically. As an example, a BO algorithm using GP-UCB and a Gaussian kernel as its covariance function $\kappa$ is no-regret since its cumulative regret $R_T \in \tilde{\mathcal{O}}\left(\sqrt{T \log^{D+1} T}\right)$ (Srinivas et al., 2012), where $\tilde{\mathcal{O}}$ denotes asymptotics up to poly-logarithmic factors. Equivalently, the average regret $R_T/T$ is in $\tilde{\mathcal{O}}\left(\sqrt{(\log^{D+1} T)/T}\right)$.

## 3 COMPUTING THE MLE USING BAYESIAN OPTIMIZATION

### 3.1 PROBLEM FORMULATION

Given an i.i.d. sample $(X_1, \cdots, X_n)$ from a finite mixture distribution $\Gamma$ with $k$ components belonging to a parametric family $\mathcal{F}$, our goal is to use a BO algorithm to find the parameters $\theta^* \in \tilde{\Theta}$ that maximize the likelihood $L$ given in (2). Formally, $\theta^* = \arg\max_{\theta \in \tilde{\Theta}} L(X; \theta)$, where $\tilde{\Theta} \subset \mathbb{R}^D$ is a $D$-dimensional compact search space. In this section, we rigorously specify $\tilde{\Theta}$, the covariance function $\kappa$ and the information we leverage to reduce the problem complexity.

The search space $\tilde{\Theta} = \bigtimes_{a=1}^{k} \tilde{\Theta}_a$ is the cross-product of the search spaces $\tilde{\Theta}_a$ for parameters of the $a$-th component of the mixture distribution, for all $a \in [k]$. As described in Table 1, a $d$-dimensional elliptical distribution is defined by a $d \times d$ shape matrix $\Sigma_a$, a $d$-dimensional location vector $\mu_a$, and possibly one additional distribution-specific parameter ($\nu_a$ for Student's t or $\beta_a$ for generalized Gaussian). For skewed distributions, we also add $d$ real-valued skewness parameters $\lambda_{a1}, \cdots, \lambda_{ad}$ (see Table 2). To compute the covariance between two likelihood values $L(X, \theta)$ and $L(X, \theta')$, where $\theta, \theta' \in \tilde{\Theta}$, we use the universal Gaussian kernel defined by

$$\kappa(\theta, \theta') = \sigma^2 \exp\left(-\frac{\|\theta - \theta'\|_2^2}{2\ell^2}\right), \tag{7}$$

where the lengthscale $\ell$ is the kernel hyperparameter.

**Dimension of the Search Space.** Let $\delta \in \mathbb{Z}_+$ denote the number of distribution-specific parameters in the parametric family (such as the degree of freedom for the Student's t distribution). Because $\mu \in \mathbb{R}^d$ and $\Sigma \in \mathbb{R}^{d \times d}$, we can naively see $\tilde{\Theta}$ as a $D$-dimensional search space, where $D = k(d^2 + d + \delta)$ for elliptic distributions. Skewed distributions add a diagonal matrix $\Lambda \in \mathbb{R}^{d \times d}$ to the model, bringing us to $D = k(d^2 + 2d + \delta)$. However, the shape matrix $\Sigma$ is necessarily positive definite (PD). Using Cholesky's decomposition, one can write $\Sigma = LL^\top$, where $L$ is a $d \times d$ lower triangular matrix with nonnegative diagonal entries and only $d(d+1)/2$ nonzero coefficients. Learning $L$ instead of $\Sigma$ allows us to factor the PD constraint directly into the search space and, by doing so, to reduce the dimensionality $D$ of the search space $\tilde{\Theta}$ to

$$D_{\text{elliptic}} = \underbrace{k}_{\text{Number of clusters}} (\underbrace{d(d+1)/2}_{\text{Shape } \Sigma_i} + \underbrace{d}_{\text{Location } \mu_i} + \underbrace{\delta}_{\text{Extra parameter}})$$

$$D_{\text{skewed}} = \underbrace{k}_{\text{Number of clusters}} (\underbrace{d(d+1)/2}_{\text{Shape } \Sigma_a} + \underbrace{d}_{\text{Location } \mu_a} + \underbrace{\delta}_{\text{Extra parameter}} + \underbrace{d}_{\text{skewness parameters}}),$$

for elliptic and skewed distributions, respectively. These dimensions scale quadratically in $d$ (but only linearly in $k$). We show below how we can leverage some information about the problem to significantly speed up the search for the maximal argument $\theta^*$ of the likelihood $L$.

**Permutation Invariance of the Likelihood.** The likelihood $L$ of the mixture model is invariant up to a permutation of the elliptical distributions parameters. Formally, for any $\theta = (\theta_1, \cdots, \theta_k) \in \tilde{\Theta}$ and any permutation $\sigma \in \mathcal{S}_k$, we have $L(X, \theta) = L(X, \sigma \circ \theta)$. Encoding this symmetry into the kernel function used by a BO algorithm drastically increases its sample efficiency, as recently shown by Brown et al. (2024). To do so, we follow their recommendations and use the kernel

$$\kappa_S(\theta, \theta') = \frac{1}{k!} \sum_{\sigma \in S(k)} \kappa(\theta, \theta'_\sigma), \tag{8}$$

where $\kappa$ is defined in (7).

**Expert Knowledge.** Finally, we can easily factor prior knowledge about the clusters. As a simple example, consider the search space $\Theta_i^\mu$ for the location vector $\mu_i$ of the $i$th elliptical distribution. Without any prior knowledge, the BO algorithm must use the naive search space $\tilde{\Theta}_i^\mu = \bigtimes_{m=1}^{d} [\min_{j \in [n]} X_{jm}, \max_{j \in [n]} X_{jm}]$. These loose bounds could be refined by an expert's knowledge on $\mu_i$ to reduce the hypervolume $\text{vol}(\tilde{\Theta})$ of the search space and speed up the search for the optimal clustering $\theta^*$.

## 3.2 ALGORITHM AND PRACTICAL CONSIDERATIONS

In Algorithm 1, we use the GP-UCB acquisition function $\varphi_t(\theta) = \mu_t(\theta) + \beta_t^{1/2} \sigma_t(\theta)$ introduced by Srinivas et al. (2012) to guide the search for the global maximizer of the log-likelihood $L$. Here, $\mu_t$ and $\sigma_t^2$ are defined in (5) and (6), respectively. Using GP-UCB, we are able to provide formal guarantees about the parameters recommended by Algorithm 1 (see Section 3.3).

Now, let us discuss a few practical aspects when running Algorithm 1.

---

**Algorithm 1:** BO for Clustering in Finite Mixture Models

---

1: **Input**: number of clusters $k$, search space $\tilde{\Theta}$, horizon $T$, finite sample $X$, sequence $(\beta_t)_{t \in [T]}$.
2: Init dataset $\mathcal{D}_0 = \emptyset$
3: **for** $t \in [T]$ **do**
4:     Select $\theta_t = \arg\max_{\theta \in \tilde{\Theta}} \mu_{t-1}(\theta_t) + \beta_{t-1}^{1/2} \sigma_{t-1}(\theta_t)$
5:     Compute $y_t = L(X; \theta_t)$
6:     Update dataset $\mathcal{D}_t = \mathcal{D}_{t-1} \cup \{(\theta_t, y_t)\}$
7: **end for**
8: Return $\hat{\theta}^T = \arg\max_{\theta \in \tilde{\Theta}} \mu_T(\theta)$

---

**Computational cost.** The computational cost is dominated by the computation of (5) and (6), both of which require the inverted $t \times t$ covariance matrix $\kappa(\mathcal{D}_t, \mathcal{D}_t)$. This requires $\mathcal{O}(t^3)$ operations.

**Choosing $\beta_t$ and $T$.** For the guarantees provided in Section 3.3 to hold, $\beta_t$ should be defined as in Theorem 2 of Srinivas et al. (2012). However, in practice, this definition of $\beta_t$ leads to over-exploration of $\tilde{\Theta}$. To achieve better performance in finite time horizons $T$, we set $\beta_t = \frac{d}{2} \log(2t)$. Furthermore, the time horizon $T$ should be chosen as large as possible, since the guarantees provided in Section 3.3 are asymptotic (i.e., they hold when $T \to +\infty$).

**Recommended mixture parameters.** Algorithm 1 returns the Bayes' optimizer $\arg\max_{\theta \in \tilde{\Theta}} \mu_t(\theta)$. These are the optimal mixture parameters given the GP surrogate at time $T$. Alternatively, one could also return the best mixture parameters explored so far, which are $\theta_{t^*}$, where $t^* = \arg\max_{t \in [T]} L(X; \theta_t)$.

## 3.3 THEORETICAL GUARANTEES

In this section, we leverage the no-regret guarantee provided by the BO framework to formulate asymptotic guarantees on the recovery of $\hat{\theta}^{\mathrm{MLE}}$ by Algorithm 1. Recall that $\rho$ is a metric over the space of parameters $\Theta$. To account for the permutation of cluster labels in the recovery of the mixture parameters, we define for any $\theta, \theta' \in \Theta$

$$\|\theta - \theta'\| = \inf_{\sigma \in \mathcal{S}_k} \sum_{a=1}^{k} \rho\left(\theta_a, \theta'_{\sigma(a)}\right),$$

where the inf is taken over all the permutations $\sigma$ of $[k]$.

The following proposition establishes the convergence of the BO-based estimator to the MLE, both in parameter space and in distribution under the Wasserstein metric.

**Proposition 1.** *Let $\pi^* \in \Delta^{k-1}$ and $\theta^* \in \Theta^k$, and let $X_1, \cdots, X_n$ be an iid sample from the mixture $M(\cdot; \pi^*, \theta^*)$. Let $(\hat{\pi}^T, \hat{\theta}^T)$ be the parameters returned by Algorithm 1 on the time horizon $T$, and suppose that $\hat{\theta}^{\mathrm{MLE}}$ is uniquely defined (up to permutations). Then,*

$$\lim_{T \to \infty} \|\hat{\theta}^T - \hat{\theta}^{\mathrm{MLE}}\| = 0.$$

*Moreover, if the moment of order $r \geq 1$ of the parametric family is finite, we also have*

$$\lim_{T \to \infty} \mathbb{E}\left[W_r\left(M(\cdot; \hat{\pi}^T, \hat{\theta}^T), M(\cdot; \hat{\pi}^{\mathrm{MLE}}, \hat{\theta}^{\mathrm{MLE}})\right)\right] = 0.$$

The condition on the finiteness of the $r$-th moment is required to ensure that the Wasserstein distance is both well defined and continuous.[1] This condition is satisfied for families with sufficiently fast-decaying tails, such as generalized Gaussians. For families with polynomially decaying densities, one must ensure the decay is strong enough; for instance, in the Student's t distribution, the

---

[1]Recall that if $(\mathcal{X}, d)$ is a Polish space and $\mathcal{P}_r(\mathcal{X})$ denotes the set of probability measures with finite $r$-th moment, then $W_r$ is continuous on $\mathcal{P}_r(\mathcal{X})$; see, e.g., (Villani et al., 2008, Corollary 6.8).

degrees-of-freedom parameter $\nu$ must exceed $r$. Note however that this condition is not linked to the performance of Algorithm 1, but to the well-definiteness of the Wasserstein metric. In particular, the first statement of Proposition 1 does not require any extra condition on the moments of the family.

We now focus on the clustering obtained using a predicted $\hat{\theta}$ instead of the true mixture parameters $\theta^*$. More precisely, given estimated parameters $\hat{\theta} = (\hat{\theta}_1, \cdots, \hat{\theta}_k)$, we consider the clustering rule

$$\forall i \in [n]\colon \hat{z}_i(\hat{\theta}) \;=\; \arg\max_{a \in [k]} f(X_i; \hat{\theta}_a). \tag{9}$$

We will make the following assumption on the likelihood ratios.

**Assumption 1** (Uniform integrability of likelihood ratio). *For every $\theta, \theta' \in \Theta$, there exists a neighborhood $N_{(\theta,\theta')}$ of $(\theta, \theta')$ such that the family $\left\{ f(x; \theta) \frac{f(x;\tilde{\theta}')}{f(x;\tilde{\theta})} ; (\tilde{\theta}, \tilde{\theta}') \in N_{(\theta,\theta')} \right\}$ is uniformly integrable.*

Assumption 1 is required to ensure the convergence of integrals $\int f(x; \theta_a^*) \left( \frac{f(x;\hat{\theta}_b^T)}{f(x;\hat{\theta}_a^T)} \right)^s dx$, when $\hat{\theta}_a^T$ and $\hat{\theta}_b^T$ are sequences of estimators, converging point-wise to $\theta_a^*$ and $\theta_b^*$, respectively. These integrals naturally appear using a Chernoff bound to control the expected loss of $\hat{z}(\hat{\theta})$. Under Assumption 1, these integrals converge to $\mathrm{Chernoff}(\theta_a^*, \theta_b^*)$, which controls the optimal error rate, as established in the following proposition.

**Proposition 2.** *Suppose that, for almost every $x \in \mathcal{X}$, $\theta \mapsto f(x; \theta)$ is continuous and strictly positive. Suppose also that Assumption 1 holds. Let $\hat{z}(\hat{\theta}^T)$ be the clustering obtained using the clustering rule (9) where the sequence of estimators $\hat{\theta}^T = (\hat{\theta}_1^T, \cdots, \hat{\theta}_k^T)$ satisfy $\lim_{T \to \infty} \|\hat{\theta}^T - \theta^*\| = 0$. Then, there exists a sequence $\eta_T$ such that $\lim_T \eta_T = 0$ and*

$$\mathbb{E}\left[ \mathrm{loss}(\hat{z}^T, z^*) \right] \;\leq\; e^{-(1+\eta_T)\min_{a \neq b \in [k]} \mathrm{Chernoff}(\theta_a^*, \theta_b^*)}.$$

We recall from Dreveton et al. (2024) that $e^{-\min_{a \neq b \in [k]} \mathrm{Chernoff}(\theta_a^*, \theta_b^*)}$ characterizes the optimal asymptotic error rate, such that no algorithm can achieve a lower error rate when $n$ is large. Proposition 2 states that the classification rule (9) achieves this optimal error rate whenever the sequence of estimators $\hat{\theta}^T$ converges to $\theta^*$ (up to a permutation of the cluster labels). Hence, this result is quite general as it can be applied to any estimator (and not only the estimates obtained by BO). Moreover, Proposition 1 shows that the estimates $\hat{\theta}^T$ obtained by BO converge to the MLE estimate. Therefore, the rate at which the expected loss of the clustering rule (9) using the estimates $\hat{\theta}^T$ returned by Algorithm 1 decreases is optimal whenever the estimate obtained by the MLE are consistent. This latter condition typically requires technical conditions on the mixture, such as strong identifiability, and we refer the reader to (Nguyen, 2013; Ho & Nguyen, 2016a;b; Heinrich & Kahn, 2018).

We conclude this discussion by noting that Assumption 1 is a mild assumption, and is in particular implied by the integrability of the likelihood-ratio (easier to check in practice). In particular, we show in Lemma 6 in the Appendix that the multivariate Student's t family satisfies Assumption 1.

## 4 NUMERICAL RESULTS

In this section, we evaluate the performance of several clustering algorithms on both synthetic and real-world datasets. Specifically, we compare Lloyd's algorithm, the EM algorithm for Gaussian mixture models (GMM) and for Student's t mixture models (SMM), spectral clustering (SC), and Algorithm 1. Lloyd's, GMM, and SC are employed through their implementations in the `scikit-learn` library with default parameters; the implementation of SMM is taken from the package `student-mixture`[2] and the algorithm is described in Peel & McLachlan (2000). Finally, Algorithm 1 is implemented with BOTorch (Balandat et al., 2020), a popular BO library. The code to reproduce our simulations is available in an anonymous online repository.[3]

---

[2]Accessible at https://pypi.org/project/student-mixture/.

[3]Accessible at https://anonymous.4open.science/r/Estimation-and-Clustering-in-Finite-Mixture-Models-Bayesian-Optimization-as-an-Alternative-to-EM-5D06.

## 4.1 STUDENT'S T MIXTURE MODEL

We first consider synthetic data sets of SMM with $n = 1000$ points in $\mathbb{R}^2$ ($d = 2$) partitioned into $k = 2$ clusters of same size. We fix $\mu_1 = (2, 1)$ and $\mu_2 = (0, 2)$ and consider the scenarios:

1. $\nu_1 = \nu_2 = 2.5$ and $\Sigma_1 = \Sigma_2 = \begin{pmatrix} 1 & \sigma \\ \sigma & 1 \end{pmatrix}$, and we vary $\sigma$ from $-0.9$ to $0.9$;

2. $\Sigma_1 = \Sigma_2 = \begin{pmatrix} 1 & 0.5 \\ 0.5 & 1 \end{pmatrix}$ and we vary $\nu_1 = \nu_2 = \nu$ from 1 to 10;

3. $\Sigma_1 = I_2$ and $\nu_1 = 2$, while $\Sigma_2 = \begin{pmatrix} 1 & 0.5 \\ 0.5 & 1 \end{pmatrix}$ and we vary $\nu_2$ from 1 to 10.

The accuracy (defined as $1 - \text{loss}(z^*, \hat{z})$, where $z^*$ and $\hat{z}$ are the true and the predicted cluster memberships, respectively) and the Wasserstein distance of order 2 between the true and the estimated mixture obtained by each algorithms for each scenario are given in Figure 1 and Figure 2, respectively.

We find that fitting a SMM with BO consistently outperforms competing methods, both in terms of clustering accuracy and parameter estimation. In contrast, EM-based approaches tailored to SMM often break down when the Student's t components are heavy-tailed.

Specifically, in the first scenario, the Student's t distribution has a small degrees-of-freedom parameter $\nu$, making the Gaussian approximation invalid (recall that as $\nu \to \infty$, the Student's t distribution converges to a Gaussian). In this regime, an EM algorithm fitting a GMM performs poorly. Simpler methods such as $k$-means can sometimes recover clusters, particularly when the components are isotropic.[4] In the second scenario, fitting a GMM works well when $\nu$ is sufficiently large, but fails otherwise. The third scenario combines features of the previous two: the first Student component is isotropic with a small $\nu_1$, while the second is anisotropic with $\nu_2$ varying. Here, we observe that the presence of even a single non-Gaussian component is enough to cause EM (GMM) to fail.

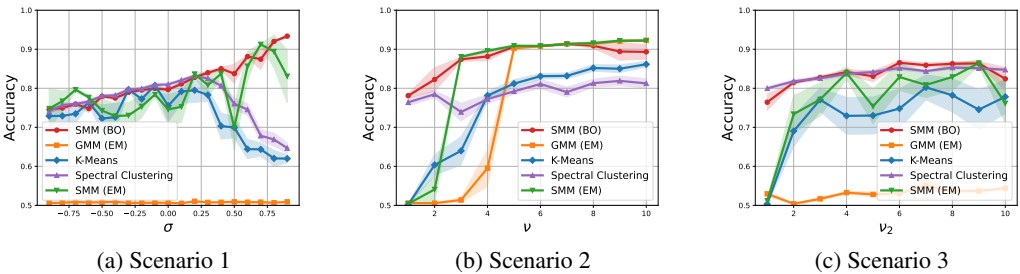

(a) Scenario 1          (b) Scenario 2          (c) Scenario 3

Figure 1: Performance of the different algorithms for clustering SMMs. Results show the clustering accuracy, averaged over 10 realizations, and error bars show the standard errors.

## 4.2 SKEWED STUDENT'S T MIXTURE MODEL

Next, we consider a setting of a mixture of a skewed and a non-skewed Student's t distribution. Both distributions have shape $\Sigma = \begin{pmatrix} 1 & 0.9 \\ 0, 9 & 1 \end{pmatrix}$, degree-of-freedom $\nu = 2.5$ and respective locations $\mu_1 = (0, 0)$ and $\mu_2 = (2, 2)$. The first distribution has a skewness vector $(\lambda, \lambda)$, where $\lambda$ varies from $-10$ to 10, while the second distribution is non-skewed.

Figure 3 demonstrates that BO consistently delivers the best performance, both in clustering accuracy and in Wasserstein distance. This advantage holds across nearly all values of $\lambda$, including the challenging regime $\lambda \in [0, 3]$ where the clusters are highly non-separable. Spectral clustering also

---

[4]As shown in Chen & Zhang (2024), $k$-means achieves optimal clustering for isotropic GMMs but is suboptimal in the anisotropic setting. Our experiments confirm this limitation, as $k$-means fails to recover anisotropic mixtures. Nonetheless, it retains some effectiveness when clustering heavy-tailed but isotropic mixtures.

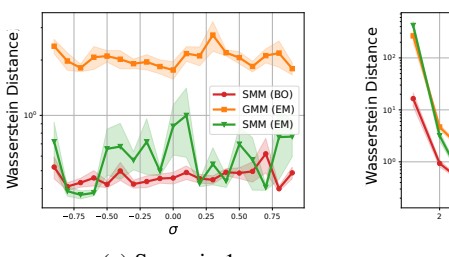 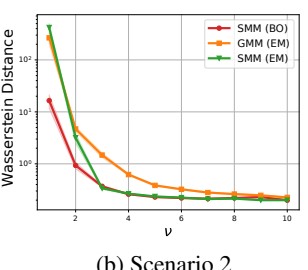 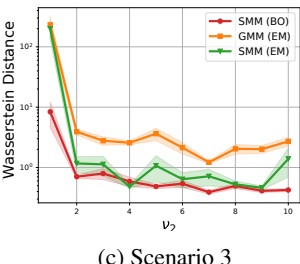

        (a) Scenario 1             (b) Scenario 2             (c) Scenario 3

Figure 2: Performance of the different algorithms for estimating SMMs. Results show the Wasserstein distance, averaged over 10 realizations, and error bars show the standard errors.

shows robust behavior when the clusters are separable, reliably recovering the clusters, whereas the remaining methods exhibit more erratic performance, achieving competitive results only in narrow parameter ranges.

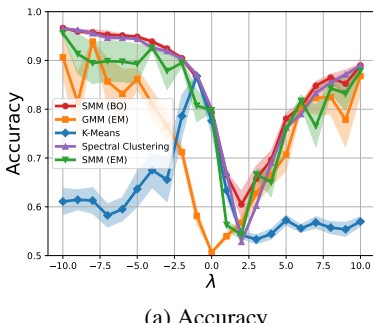 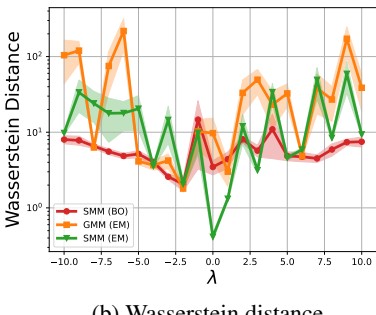

           (a) Accuracy                   (b) Wasserstein distance

Figure 3: Performance of the different algorithms for estimating mixtures with skewed components. Results show the (a) accuracy and (b) the Wasserstein distance, averaged over 10 realizations, and error bars show the standard errors.

## 5 CONCLUSION

In this work, we proposed a BO algorithm (Section 3.2) as an alternative to the EM algorithm for MLE and clustering in finite mixtures of elliptical distributions. Theoretically, we established that the sequence of BO estimates converges to the MLE up to label permutation, and that the resulting clustering achieves asymptotically the optimal misclassification rate under mild regularity assumptions (Section 3.3). To the best of our knowledge, these constitute the first global convergence guarantees for a practically implementable algorithm in this setting. Empirically, BO consistently outperforms EM, Lloyd's algorithm, and spectral clustering on challenging synthetic Student's $t$ mixtures, particularly in heavy-tailed and anisotropic regimes where standard methods are known to fail (Section 4.1). Finally, we showed that the BO framework extends naturally to broader clustering tasks, as illustrated by its strong performance on skewed Student's $t$ mixtures (Section 4.2).

Beyond the results presented here, the versatility of the BO framework may be used to address other challenging clustering problems. As an example, online clustering (i.e., datapoints come sequentially as described in Cohen-Addad et al. (2021)), remains a challenging task because the optimized likelihood $L_{X_t}$ changes constantly with the online sample $X_t$. Time-varying Bayesian Optimization (TVBO) can account for time-varying objective functions and successfully optimize them in an online fashion (Bardou et al., 2024). We keep the time-varying extension of the method described in this paper as a future work.

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

# A  PROOFS

## A.1  PROOF OF PROPOSITION 1

*Proof.* Denote by $L(X; \pi, \theta) = \sum_i \log M(X_i; \pi, \theta)$ the likelihood function of the sample $X_1, \cdots, X_n$ with respect to the mixture parameters $\pi, \theta$. Srinivas et al. (2012, Theorem 2) ensures that $R_T = \sum_{t=1}^T L(X; \hat{\pi}^{\mathrm{MLE}}, \hat{\theta}^{\mathrm{MLE}}) - L(X; \hat{\pi}_t, \hat{\theta}_t) \in \mathcal{O}\left(\sqrt{T \log^{D+1}(T)}\right)$ with high probability (w.h.p.). Because the MLE is uniquely defined (up to permutations), the sub-linearity of $R_T$ ensures that the estimation $\hat{\theta}^T$ returned by Algorithm 1 defined by $\hat{\theta}^T = \arg\max_{\theta \in \tilde{\Theta}} \mu_T(\theta)$ satisfies $\hat{\theta}^T \to \hat{\theta}^{\mathrm{MLE}}$ up to permutations.[5] More precisely, there exists a sequence $(\sigma_T)_{T \in \mathbb{Z}_+}$ of permutations such that

$$\lim_{T \to \infty} \sum_{a=1}^k \rho\left(\hat{\theta}_a^T, \hat{\theta}_{\sigma_T(a)}^{\mathrm{MLE}}\right) = 0.$$

Let $r \geq 1$ be such that the $r$-th moment of $M$ is finite. By continuity of the Wasserstein metric (see for example Villani et al. (2008, Corollary 6.8)), the convergence $\sigma_T \circ \hat{\theta}^T \to \hat{\theta}^{\mathrm{MLE}}$ for some sequence of permutations $(\sigma_T)$ established above implies that

$$\lim_{T \to \infty} W_r\left(M(\cdot; \sigma_T \circ \pi, \sigma_T \circ \theta), M(\cdot; \hat{\pi}^{\mathrm{MLE}}, \hat{\theta}^{\mathrm{MLE}})\right) = 0.$$

Moreover, because the distribution $M$ is permutation-invariant (that is, $M(\cdot; \pi, \theta) = M(\cdot; \sigma \circ \pi, \sigma \circ \theta)$ for any permutation $\sigma$), we obtain

$$\lim_{T \to \infty} W_r\left(M(\cdot; \hat{\pi}^T, \hat{\theta}^T), M(\cdot; \hat{\pi}^{\mathrm{MLE}}, \hat{\theta}^{\mathrm{MLE}})\right) = 0.$$

In other words, the mixture distribution defined by the BO estimates converges, in Wasserstein distance, to the MLE mixture distribution, without the need to relabel the components.

Finally, since $\tilde{\Theta}$ is compact, the continuity of the Wasserstein metric ensures that $\sup_{\theta, \theta' \in \tilde{\Theta}} W_r\left(M(\cdot; \pi, \theta), M(\cdot; \pi', \theta')\right) < \infty$. Therefore, by the dominated convergence theorem,

$$\lim_{T \to \infty} \mathbb{E}\left[W_r\left(M(\cdot; \hat{\pi}^T, \hat{\theta}^T), M(\cdot; \hat{\pi}^{\mathrm{MLE}}, \hat{\theta}^{\mathrm{MLE}})\right)\right] = \mathbb{E}\left[\lim_{T \to \infty} W_r\left(M(\cdot; \hat{\pi}^T, \hat{\theta}^T), M(\cdot; \hat{\pi}^{\mathrm{MLE}}, \hat{\theta}^{\mathrm{MLE}})\right)\right]$$
$$= 0.$$

In other words, the expected Wasserstein distance between the estimated mixture distribution and the MLE distribution also vanishes asymptotically. □

## A.2  PROOF OF PROPOSITION 2

*Proof.* Denote by $\sigma$ the optimal permutation, that is, $\sigma_T = \arg\min_{\sigma \in \mathcal{S}_k} \|\theta^* - \hat{\theta}^T\|$. Without loss of generality, we suppose that $\sigma_T$ is the identity. Equation (3) yields that

$$\mathbb{E}\left[\mathrm{loss}(\hat{z}^T, z^*)\right] \leq \frac{1}{n} \mathbb{E}\left[\sum_{i=1}^n \mathbb{1}(\hat{z}_i^T \neq z_i^*)\right] = \frac{1}{n} \sum_{i=1}^n \mathbb{P}\left[\hat{z}_i^T \neq z_i^*\right].$$

---

[5] A sublinear $R_T$ does not ensure that $r_t = L(X; \hat{\pi}^{\mathrm{MLE}}, \hat{\theta}^{\mathrm{MLE}}) - L(X; \hat{\pi}_t, \hat{\theta}_t) \to 0$ point-wise. However, the Bayes' optimizer $\hat{\theta}^T$ returned by Algorithm 1 at time horizon $T$, that is $\hat{\theta}^T = \arg\max_{\theta \in \tilde{\Theta}} \mu_T(\theta)$ does satisfy $L(\hat{\theta}^{\mathrm{MLE}}) - L(\hat{\theta}^T) \to 0$ or, equivalently, $\hat{\theta}^T \to \hat{\theta}^{\mathrm{MLE}}$.

Fix $i \in [n]$ and observe that

$$
\left\{ \hat{z}_i^T \neq z_i^* \right\} = \left\{ z_i^* \notin \underset{b \in [k]}{\arg\max} \, f(X_i; \hat{\theta}_b^T) \right\}
$$

$$
= \left\{ \exists b \in [k] \setminus \{z_i^*\} \colon f(X_i; \hat{\theta}_b^T) > f(X_i; \hat{\theta}_{z_i^*}^T) \right\}.
$$

Denote by $a = z_i^*$ the true cluster of $i$. By a union bound,

$$
\mathbb{P}\left[ \hat{z}_i^T \neq z_i^* \right] \leq \sum_{b \in [k] \setminus \{a\}} \mathbb{P}\left( f(X_i; \hat{\theta}_b^T) > f(X_i; \hat{\theta}_a^T) \right)
$$

$$
\leq (k-1) \max_{b \in [k] \setminus \{a\}} \mathbb{P}\left( f(X_i; \hat{\theta}_b^T) > f(X_i; \hat{\theta}_a^T) \right).
$$

Denote $\mathbb{P}_{\theta_a^*}(\cdot) = \mathbb{P}(\cdot; \theta_a^*)$ and observe that

$$
\mathbb{P}\left( f(X_i; \hat{\theta}_b^T) > f(X_i; \hat{\theta}_a^T) \right) = \mathbb{P}_{\theta_a^*}\left( f(X; \hat{\theta}_b^T) > f(X; \hat{\theta}_a^T) \right).
$$

Therefore,

$$
\mathbb{E}\left[ \mathrm{loss}(\hat{z}, z^*) \right] \leq (k-1) \max_{b \in [k] \setminus \{a\}} \mathbb{P}_{\theta_a^*}\left( f(X; \hat{\theta}_b^T) > f(X; \hat{\theta}_a^T) \right). \tag{10}
$$

We have, for any $s > 0$,

$$
\mathbb{P}_{\theta_a^*}\left( f(X; \hat{\theta}_b^T) > f(X; \hat{\theta}_a^T) \right) = \mathbb{P}_{\theta_a^*}\left( e^{s \log \frac{f(X; \hat{\theta}_b^T)}{f(X; \hat{\theta}_a^T)}} > 1 \right)
$$

$$
\leq \mathbb{E}_{\theta_a^*}\left[ e^{s \log \frac{f(X; \hat{\theta}_b^T)}{f(X; \hat{\theta}_a^T)}} \right]
$$

$$
= \int f(x; \theta_a^*) \left( \frac{f(x; \hat{\theta}_b^T)}{f(x; \hat{\theta}_a^T)} \right)^s dx. \tag{11}
$$

where the inequality follows from Markov's inequality. Therefore, by combining (10) and (11), we obtain

$$
\mathbb{E}\left[ \mathrm{loss}(\hat{z}, z^*) \right] \leq (k-1) \max_{b \in [k] \setminus \{a\}} \inf_{s \in (0,1)} \int f(x; \theta_a^*) \left( \frac{f(x; \hat{\theta}_b^T)}{f(x; \hat{\theta}_a^T)} \right)^s dx
$$

$$
= (k-1) \exp\left( \max_{b \in [k] \setminus \{a\}} \log \left( \inf_{s \in (0,1)} \int f(x; \theta_a^*) \left( \frac{f(x; \hat{\theta}_b^T)}{f(x; \hat{\theta}_a^T)} \right)^s dx \right) \right). \tag{12}
$$

Let

$$
I_{a,b}(s, T) = \int f(x; \theta_a^*) \left( \frac{f(x; \hat{\theta}_b^T)}{f(x; \hat{\theta}_a^T)} \right)^s dx \quad \text{and} \quad J_{a,b}(s) = \int \left( f(x; \theta_a^*) \right)^{1-s} \left( f(x; \theta_b^*) \right)^s dx.
$$

Furthermore, because $\theta_a^* \neq \theta_b^*$, Lemma 4 ensures that the infimum of $J_{a,b}(s)$, $\inf_{s \in [0,1]} J_{a,b}(s)$, is attained at some $t^*$ bounded away from 0 and from 1. Let $\varepsilon > 0$ and $K = [\varepsilon, 1 - \varepsilon]$ such that $t^* \in K$. Lemma 3 ensures that, for any pair $a \neq b$, we have

$$
\lim_{T \to \infty} \inf_{s \in K} I_{a,b}(s, T) = \inf_{s \in K} J_{a,b}(s). \tag{13}
$$

Combining (13) with (12), and noticing that

$$
\mathrm{Chernoff}(\theta_a^*, \theta_b^*) = -\log\left( \inf_{s \in (0,1)} J_{a,b}(s) \right),
$$

and recalling that $\inf_{s \in (0,1)} J_{a,b}(s) = \inf_{s \in K} J_{a,b}(s)$, we obtain

$$\mathbb{E}\left[\text{loss}(\hat{z}, z^*)\right] \ \leq \ (k-1)\exp\left(-(1+o(1))\min_{b \in [k] \setminus \{a\}} \text{Chernoff}(\theta_a^*, \theta_b^*)\right),$$

which concludes the proof. $\qquad\qquad\square$

### A.3 ADDITIONAL LEMMAS

**Lemma 3.** *Let* $\{f(\cdot; \theta); \theta \in \Theta\}$ *be a parametric family of pdf (over* $\mathbb{R}^d$*). Let* $\theta, \theta' \in \Theta$ *such that* $\theta \neq \theta'$*. Suppose that:*

1. *For almost every* $x$, $\tilde{\theta} \mapsto f(x; \tilde{\theta})$ *is continuous at* $\theta$ *and at* $\theta'$*;*

2. *There exists a neighborhood* $N_{(\theta, \theta')}$ *of* $(\theta, \theta')$ *such that the family of functions* $\left\{ x \mapsto f(x; \theta)\left(\frac{f(x; \tilde{\theta})}{f(x; \tilde{\theta}')}\right); (\tilde{\theta}, \tilde{\theta}') \in N_{(\theta, \theta')} \right\}$ *is uniformly integrable.*

*Then, for any sequence* $(\theta_T)_{T \in \mathbb{Z}_+}$ *and* $(\theta'_T)_{T \in \mathbb{Z}_+}$ *such that* $\lim_{T \to \infty} \theta_T = \theta$ *and* $\lim_{T \to \infty} \theta'_T = \theta'$ *and for any compact* $K \subseteq [0, 1]$*, we have*

$$\lim_{T \to \infty} \inf_{s \in K} \int f(x; \theta)\left(\frac{f(x; \theta'_T)}{f(x; \theta_T)}\right)^s dx \ = \ \inf_{s \in K} \int \left(f(x; \theta)\right)^{1-s}\left(f(x; \theta')\right)^s dx.$$

*Proof.* Denote

$$I_T(s) \ = \ \int f(x; \theta)\left(\frac{f(x; \theta'_T)}{f(x; \theta_T)}\right)^s dx \quad \text{and} \quad J(s) \ = \ \int \left(f(x; \theta)\right)^{1-s}\left(f(x; \theta')\right)^s dx.$$

The proof follows two steps. We first establish the point-wise convergence of the sequence of functions $(I_T(\cdot))_{T \in \mathbb{Z}_+}$ to the function $J(\cdot)$ using the uniform integrability assumption. Next we refine it to an uniform convergence using the convexity of each function $I_T(\cdot)$.

(i) *Point-wise convergence in* $s$: $\lim_T I(s, T) = J(s)$. Fix $s \in (0, 1)$. For almost every $x \in \mathcal{X}$, the continuity of $\theta \mapsto f(x; \theta)$ and the convergence of $\theta_T \to \theta$ and $\theta'_T \to \theta'$ imply that

$$\frac{f(x; \theta'_T)}{f(x; \theta_T)} \ \to \ \frac{f(x; \theta')}{f(x; \theta)}.$$

Hence, the integrand $f(x; \theta)\left(\frac{f(x; \theta'_T)}{f(x; \theta_T)}\right)^s$ of $I_T(s)$ converges point-wise to the integrand $\left(f(x; \theta)\right)^{1-s}\left(f(x; \theta')\right)^s$ of $J(s)$.

Denote by $N(\theta, \theta')$ the neighborhood appearing in Assumption 1. Observe that, for $t$ large enough (say, $t \geq T_1$ for some $T_1 > 0$), we have $(\theta_T, \theta'_T) \in N(\theta, \theta')$. Moreover, using the inequality $u^s \leq 1 + u$ valid for all $u \geq 0$, we have

$$f(x; \theta)\left(\frac{f(x; \tilde{\theta})}{f(x; \tilde{\theta}')}\right)^s \ \leq \ f(x; \theta) + f(x; \theta)\left(\frac{f(x; \tilde{\theta})}{f(x; \tilde{\theta}')}\right),$$

ensuring the uniform integrability of the family $\{f(x; \theta)\left(\frac{f(x; \tilde{\theta})}{f(x; \tilde{\theta}')}\right)^s; (\tilde{\theta}, \tilde{\theta}') \in N_{(\theta, \theta')}\}$ for any $s \in [0, 1]$.

Hence, the family $\{f(x; \theta)\left(\frac{f(x; \theta'_T)}{f(x; \theta_T)}\right)^s; t \geq T_1\}$ is uniformly integrable. Vitali's theorem therefore implies that

$$\lim_{T \to \infty} I_T(s) \ = \ J(s)$$

for each fixed $s \in (0, 1)$.

(ii) *Convexity and uniform convergence on compacts.* For any $T \in \mathbb{Z}_+$, the functions $s \mapsto I_T(s)$ are convex. Indeed, $I_T(s) = \mathbb{E}_\theta \left[ e^{s \log r_T(X)} \right]$ is the moment generating function (MGF) of some random variable, with $r_T(X) = \frac{f(X;\theta_T')}{f(X;\theta_T)}$. The pointwise convergence $I_T(s) \to J(s)$ for all $s \in [0,1]$ together with convexity implies uniform convergence on every compact sub-interval of $[0,1]$. In particular, for the compact $K$ introduced earlier, we have

$$\lim_{T \to \infty} \sup_{s \in K} |I(s,T) - J(s)| = 0.$$

Because uniform convergence on a compact implies the convergence of the minimum, we finally obtain

$$\lim_{T \to \infty} \inf_{s \in K} I(s,T) = \inf_{s \in K} J(s).$$

$\square$

**Lemma 4.** *Let $f, g$ be two distinct probability densities. The infimum of $\inf_{s \in [0,1]} \int f^s g^{1-s}$ is attained at some $s^* \in (0,1)$.*

*Proof.* Let $\varphi(s) = \int f^s g^{1-s}$. Observe that, by standard arguments, $\varphi$ is continuous on $[0,1]$. Moreover, $\varphi(0) = \varphi(1) = 1$. For all $s \in (0,1)$, Holder's inequality implies that $\int f^s g^{1-s} \le \left( \int f \right)^s \left( \int g \right)^{1-s}$ and therefore, as $\int f = \int g = 1$ because $f$ and $g$ are pdfs,

$$\varphi(s) \le 1 \quad \text{for all } s \in (0,1).$$

Equality in Hölder occurs only in the degenerate case $f = cg$ for some constant $c$. The normalization condition on $f$ and $g$ imposes $c = 1$ and thus $f = g$ almost everywhere. Consequently, for $f \ne g$ we have

$$\varphi(s) < 1 \quad \text{for all } s \in (0,1).$$

Thus $\varphi$ takes the value 1 at the endpoints and values strictly smaller than 1 inside, so that the minimum over $[0,1]$ (and hence also the infimum over $(0,1)$) is strictly smaller than 1 and cannot occur at the endpoints. $\square$

### A.4 Uniform integrability of likelihood ratio for specific families

The purpose of this section is to establish that the Student's t family satisfy Assumption 1.

First, we recall that the uniform integrability of a family of random variables is typically verified thorough a slightly stronger, but easier to check, condition, which is the boundedness of the moment of order $1 + \delta$ for some $\delta > 0$. In particular, the following lemma is a direct application of standard properties regarding uniform integrability.

**Lemma 5** (Uniform $L^{1+\delta}$ likelihood-ratio condition implies Assumption 1)**.** *Let $\delta > 0$. For every $\theta, \theta' \in \Theta$, there exists a neighborhood $N_{(\theta,\theta')}$ of $(\theta, \theta')$ such that*

$$\sup_{(\tilde\theta,\tilde\theta') \in N_{(\theta,\theta')}} \mathbb{E}_\theta \left[ \left( \frac{f(x;\tilde\theta')}{f(x;\tilde\theta)} \right)^{1+\delta} \right] < \infty.$$

*Then the family $\mathcal{F} = \{f(\cdot;\theta); \theta \in \Theta\}$ satisfies Assumption 1.*

For a symmetric matrix $\Sigma$, we denote $eigenmin(\Sigma)$ (resp., $eigenmax(\Sigma)$) its smallest (resp., largest) eigenvalue. The following lemma ensures that the family of multivariate Student's t distributions satisfies Assumption 1.

**Lemma 6.** *Let $\mathcal{F} = \{f(x;\nu,\mu,\Sigma) = \frac{\Gamma(\frac{\nu+d}{2})}{(\nu\pi)^{d/2}\Gamma(\frac{\nu}{2})|\Sigma|^{1/2}} \cdot \left( 1 + \frac{(x-\mu)^T \Sigma^{-1}(x-\mu)}{\nu} \right)^{-\frac{\nu+d}{2}} ; \nu > 0, \mu \in \mathbb{R}^d, \Sigma \in \mathcal{S}_d^{++} \}$ be the family of Student's t distributions. Let $\Theta = \mathbb{R}_+^* \times \mathbb{R}^d \times \mathcal{S}_d^{++}$, and let $\tilde\Theta \subset \Theta$*

*be compact. Assume that for all $(\nu, \mu, \Sigma) \in \tilde{\Theta}$,*

$$\nu_{\min} = \inf_{(\nu,\mu,\Sigma)\in\tilde{\Theta}} \nu > 0 \quad and \quad 0 < \lambda_{\min} \leq eigenmin(\Sigma) \leq eigenmax(\Sigma) \leq \lambda_{\max} < \infty,$$

*for some constants $\nu_{\min} > 0$ and $0 < \lambda_{\min} \leq \lambda_{\max} < \infty$. Then, for every $\theta, \theta' \in \Theta$, there exists a $\delta > 0$ and a neighborhood $N_{(\theta,\theta')}$ of $(\theta, \theta') \in \Theta \times \Theta$ such that*

$$\sup_{(\tilde{\theta},\tilde{\theta}')\in N_{(\theta,\theta')}} \mathbb{E}_\theta \left[ \left( \frac{f(x;\tilde{\theta}')}{f(x;\tilde{\theta})} \right)^{1+\delta} \right] < \infty.$$

*In particular, the family $\mathcal{F}$ satisfies Assumption 1.*

Note that restricting the parameters to belong in the compact set $\tilde{\Theta}$ ensuring that the likelihood remains bounded over $\tilde{\Theta}$ because $\nu$ is bounded away from $0$ and the scale matrices $\Sigma$ are uniformly well-conditioned.

*Proof.* Fix $\theta = (\nu, \mu, \Sigma) \in K$ and $\theta' = (\nu', \mu', \Sigma') \in K$.

Observe that there exists constants $C_1, C_2, \alpha_1, \alpha_2 > 0$ such that for all $x \in \mathbb{R}^d$

$$C_1 \left(1 + \alpha_1 \|x - \mu\|^2\right)^{-(\nu+d)/2} \leq f(x;\theta) \leq C_2 \left(1 + \alpha_2 \|x - \mu\|^2\right)^{-(\nu+d)/2}.$$

Hence,

$$f(x;\theta) \asymp \|x\|^{-(\nu+d)} \quad when \quad \|x\| \to \infty.$$

Consider the quantity

$$I(x;\theta,\tilde{\theta},\tilde{\theta}') = f(x;\theta) \left( \frac{f(x;\tilde{\theta}')}{f(x;\tilde{\theta})} \right)^{1+\delta},$$

where $\tilde{\theta} = (\tilde{\nu}, \tilde{\mu}, \tilde{\Sigma}) \in K$ and $\theta' = (\tilde{\nu}', \tilde{\mu}', \tilde{\Sigma}') \in K$ belong to a neighborhood of $\theta$ and $\theta'$. Observe that, for large $\|x\|$,

$$I(x;\theta,\tilde{\theta},\tilde{\theta}') \leq C\|x\|^{-(\nu+d)} \left( \frac{\|x\|^{-(\tilde{\nu}'+d)}}{\|x\|^{-(\tilde{\nu}+d)}} \right)^{1+\delta}$$

$$\leq C\|x\|^{-(\nu+d+(1+\delta)(\tilde{\nu}'-\tilde{\nu}))}.$$

Because the integral $\int \|x\|^{-p}$ is finite iff $p > d$, we require $\nu + (1+\delta)(\tilde{\nu}' - \tilde{\nu}) > 0$, or equivalently

$$\nu > (1+\delta)(\tilde{\nu} - \tilde{\nu}'). \tag{14}$$

*Case (i): $\nu \leq \nu'$.* Because we restrict $\tilde{\theta}$ and $\tilde{\theta}'$ to be in a neighborhood of $\theta, \theta'$, we can shrink the neighborhood $N$ so that $\tilde{\nu} \leq \tilde{\nu}'$ for all element in $\tilde{\theta}, \tilde{\theta}' \in N$, and the condition (14) is satisfied for an arbitrary $\delta$.

*Case (ii): $\nu > \nu'$.*

In that case we can choose the neighborhood so that

$$|\tilde{\nu} - \nu| \leq \varepsilon \quad and \quad |\tilde{\nu}' - \nu'| \leq \varepsilon,$$

where $\varepsilon$ is small enough (we will see that imposing $\varepsilon < \nu'/2$ is enough). The choice of this neighborhood ensures that $\tilde{\nu} - \tilde{\nu}' \leq \nu - \nu' + 2\epsilon$. The condition (14) becomes equivalent to

$$\nu > (1+\delta)(\nu - \nu') + (1+\delta)2\epsilon,$$

which can be recast in

$$\delta < \frac{\nu' - 2\epsilon}{\nu - \nu' + 2\epsilon}.$$

We observe that, playing on $\epsilon$, we can establish the finiteness of the $1 + \delta$ moment for all $\delta < \frac{\nu'}{\nu-\nu'}$. $\qquad\square$

# B  COMMON ELLIPTIC DISTRIBUTIONS

Tables 1 and 2 summarize some common non-skewed and skewed parametric distributions.

Table 1: Parametric families $\mathcal{F}$ of elliptic distributions considered in this work. Each family involves a location parameter $\mu \in \mathbb{R}^d$, a scale matrix $\Sigma \in \mathcal{S}_{++}^d$, and potentially other real-valued parameters (a degree of freedom $\nu$ for the Student's t-distribution, and a shape $\beta$ for the Generalized Gaussian). The densities are given by $f(x; \theta) = \frac{C}{|\Sigma|^{1/2}} \cdot g(u)$, where $u = (x - \mu)^\top \Sigma^{-1}(x - \mu)$, where $g \colon \mathbb{R}_+ \to \mathbb{R}$ is the generator function and $C$ is the corresponding normalization constant.

| Name | Parameters $\Theta$ | Density Generator $g$ | Normalization constant $C$ |
|---|---|---|---|
| Gaussian | $\mu \in \mathbb{R}^d, \Sigma \in \mathcal{S}_{++}^d$ | $\exp\left(-\frac{1}{2}u\right)$ | $(2\pi)^{-\frac{d}{2}}$ |
| Student's t | $\mu \in \mathbb{R}^d, \Sigma \in \mathcal{S}_{++}^d, \nu > 0$ | $\left(1 + \frac{u}{\nu}\right)^{-\frac{\nu+d}{2}}$ | $\frac{\Gamma(\frac{\nu+d}{2})}{(\nu\pi)^{d/2}\Gamma(\frac{\nu}{2})}$ |
| Gen. Gaussian | $\mu \in \mathbb{R}^d, \Sigma \in \mathcal{S}_{++}^d, \beta > 0$ | $\exp\left(-\frac{1}{2}u^\beta\right)$ | $\frac{\beta\Gamma(\frac{d}{2})}{(2^{1/\beta}\pi)^{d/2}\Gamma(\frac{d}{2\beta})}$ |

Table 2: Parametric families $\mathcal{F}$ of multivariate skewed distributions. Each family extends a corresponding non-skewed distribution by incorporating a skewness vector $\Lambda = (\lambda_1, \cdots, \lambda_d) \in \mathbb{R}^d$. In the table, $\phi(\cdot; \mu, \Sigma)$ denote the pdf of a Gaussian distribution with mean $\mu$ and covariance $\Sigma$, while $\Phi(\cdot)$ is the univariate standard normal cdf. Similarly, $t_\nu(\cdot; \mu, \Sigma)$ is the pdf of a $t$-distribution with degree of freedom $\nu$, location $\mu$, and shape $\Sigma$, while $T_\nu(\cdot)$ is the univariate Student's t cdf (with degree of freedom $\nu$). Finally, we let $q(x) = (x - \mu)^T \Omega^{-1}(x - \mu)$.

| Name | Parameters $\Theta$ | pdf $f(x)$ |
|---|---|---|
| Skewed normal | $\mu, \Sigma, \Lambda$ | $2\phi(x - \mu; 0_d, \Sigma)\, \Phi(\Lambda^T \Sigma^{-1/2}(x - \mu) \,|\, 0_d, \Delta)$ |
| Skewed Student's t | $\nu, \mu, \Sigma, \Lambda$ | $2t_\nu(x - \mu; 0_d, \Sigma)\, T_{\nu+d}\left(\sqrt{\frac{\nu+d}{\nu+q(x)}} \Lambda^T \Sigma^{-1/2}(x - \mu)\right)$ |

