# OpenReview forum: "Estimation and Clustering in Finite Mixture Models: Bayesian Optimization as an Alternative to EM"
_ICLR.cc/2026/Conference — Submitted to ICLR 2026_

### Official Review · Reviewer_dACm · 2025-10-28

**Soundness:** 3
**Presentation:** 2
**Contribution:** 2
**Rating:** 2
**Confidence:** 5

**Summary:**

This paper proposes a new approach for parameter estimation in elliptical finite mixture models by replacing the traditional Expectation–Maximization (EM) algorithm with Bayesian Optimization (BO). The idea is to train a Gaussian Process (GP) model that takes as input the mixture model parameters and provides as output the log-likelihood of the model for a given dataset.  At each iteration the trained GP model is used to select new candidate parameters using a GP-UCB acquisition function and the new training point is added to the GP training set,
thus GP training is repeated with the augmented training set. The authors provide theoretical results such as a proof that the proposed BO framework achieves global convergence to the maximum likelihood estimate (MLE). Experiments are performed on two synthetic datasets involving 2d Student’s t and skewed Student’s t mixtures of two components.

**Strengths:**

S1. A new approach is presented for mixture model training based on Bayesian Optimization.
S2. It is shown that the method converges to the MLE.

**Weaknesses:**

W1. The method is theoretically sound, but difficult to apply even for moderate values of d, since the number of parameters is of d^2 order.
W2. There are several hyperparameters to be set.
W3. There are several unclear points regarding method implementation (see questions below)
W4. Experiments using only two synthetic datasets of very low dimension (2d) and with only two Student-t components are not sufficient.

**Questions:**

Q1. There is no detail on how steps (4) and (8) of Algorithm 1 are implemented and what is their complexity.
Q2. In line 78 it is mentioned that experiments on real datasets have been conducted, but I cannot find those results in the paper.
Q3. In addition to the omission of experiments on real data, the reported experimental evaluation is completely insufficient
(two datasets with two component mixtures on two-dimensional data).
Q4. What is the execution time of the method compared to SMM? How is the SMM initialized? Is it executed only once or from several initializations keeping the solution with best likelihood?
Q5. It seems that priors \pi_i are not included in the parameters.
Q6. What is the execution time to run the method?
Q7. It seems expensive to apply the method even for moderate values of input dimension d (e.g d=10).
Q7. How did you select the number of iterations T and other method hyperparameters?
Q8. Only the comparison with SMM is meaningful in the experiments.
Q9. What about clustering performance? NMI values of the obtained solutions with respect to the ground truth solution should be provided.
Q10. It would add value to the paper if plots of the 2d datasets were provided.
Q11. How do you select the initial training points and specify their number?

---

### Official Review · Reviewer_8NNN · 2025-10-30

**Soundness:** 2
**Presentation:** 2
**Contribution:** 2
**Rating:** 2
**Confidence:** 4

**Summary:**

The paper describes using Bayesian Optimization (BO) to learn the parameters of a mixture model. Since BO is a black box optimization method, it comes with a theoretical guarantee that the global optimum will be found with an infinite number of steps. Experiments show that with a finite number of steps, the approach often outperforms EM-GMM (when the data is not generated by a GMM).

**Strengths:**

Optimizing the log likelihood of a mixture model is a fundamental problem in ML and the approach provided here is original. It is certainly interesting to see how well BO will work in this setting.

**Weaknesses:**

The paper claims to establish global optimality guarantees, but these guarantees only hold when the number of evaluations T goes to infinity. I believe that this weakens the claim that  the paper provides the " first global convergence
guarantees for a practically implementable algorithm in this setting". Allowing an infinite number of evaluations is not practically implementable.

In practice, we expect the number of evaluations to grow exponentially with the dimensionality of the data, and indeed all the experiments in the paper are with two dimensional data. For such a setting, even simple grid search followed by a few iterations of EM might work.

The experiments make no mention of run times or the initialization procedure for EM. Presumably by trying more random restarts the EM results will improve and this is also true for the BO approach. So a fair comparison should take into account run times as well.

Finally, the experimental results indeed show consistent failures of EM-GMM but this is not particularly surprising since the data was not generated by a GMM. The EM-SMM results are mostly comparable to those of BO except for a few datapoints in figures 2b,2c.

**Questions:**

What are the runtimes?

Have you tried different initialisation strategies for EM?

How does BO perform in higher dimensions?

---

### Official Review · Reviewer_1HXx · 2025-11-04

**Soundness:** 2
**Presentation:** 3
**Contribution:** 2
**Rating:** 6
**Confidence:** 3

**Summary:**

The paper investigates the use of Bayesian optimization (BO) for computing the maximum likelihood estimate (MLE) of general elliptical mixture models, given that standard alternatives such as EM are known to suffer from local optima. It establishes that the BO estimates converge to the true MLE, which gives asymptotic global convergence guarantees.Furthermore, under certain technical assumptions, the clustering error achieved by BO converges to the optimal misclassification error. An experimental study on synthetic and real datasets confirms the superior performance of BO compared to EM.

**Strengths:**

- Even though the main idea and the corresponding algorithm are quite simple, the authors are able to show global convergence properties, which the main competitors (EM) generally lack for arbitrary elliptical distributions. This could be significant in situations where EM gets stuck in local optima.
- The paper is well written and the exposition is detailed while the theory seems sound (even though I did not check all proofs in detail).
- The experimental study shows that BO achieves the most consistent performance across a wide range of parameterizations, unlike EM which can heavily depend on the exact parameters.

**Weaknesses:**

- To the best of my understanding, the paper did not address the question of execution time and/or sample complexity. It is true that EM may get stuck to local minima - but how does BO compare to EM in terms of execution time? Actually, as the experimental figures show, EM can often achieve very good performance and comparable to BO. It is then natural to ask how the two frameworks compare in terms of total execution time. Obviously, if BO has a huge budget, it should be able to perform better than a gradient-based method that can get stuck in local minima. But if the extra time cost is very high, this could be an important consideration.
- The authors experimented with Student's T mixture model. I was not clear why they did not use more choices. The Gaussian mixture model is presumably less challenging? But what about other models following the elliptical distribution? I feel a more diverse set of mixture models would paint a more accurate picture of the relative performance of the two frameworks. The results on Student's T mixture model are definitely interesting and encouraging, but they could have ben complemented by results from other distributions, too.

**Questions:**

- Can the authors elaborate on the execution time of BO, and how it compares to EM? They already mention various regret bounds in Section 2, but these are mostly asymptotic. What about the total execution time, especially with respect to EM, especially in the challenging settings/parameterizations?
- Perhaps the authors could also include convergence plots showing how BO reaches the MLE over time?
- In the cases where EM performed poorly, does EM benefit from multiple runs with different random initialisations? I was wondering whether simple workarounds might help improve EM's performance (irrespective of how BO performs).
- Why have the authors only experimented with Student's T mixture model? Why not with Gaussian mixture models and/or other elliptical distributions? I feel this would give a more accurate picture of the relative strength of the proposed method.

---

### Official Review · Reviewer_LKcJ · 2025-11-04

**Soundness:** 2
**Presentation:** 4
**Contribution:** 3
**Rating:** 4
**Confidence:** 4

**Summary:**

The authors address the problem of finding the parameters of a mixture
model. Contrary to most existing methods, the contribution provides a
global optimum by leveraging the Bayesian optimization framework. An
application of this framework to mixture is described along with
theoretical guaranties. Toy examples assess the efficiency of the
proposed method.

**Strengths:**

Most of methods for the estimation of the parameters of a mixture
model are limited to local maxima and may be trapped into arbitrarly
bad extrema. It thus of primary importance to study methods able to
provide theoretical guaranties on the quality of the output (here a
global maximum). This is even more interesting for mixtures of
distributions other than Gaussians which are more difficult to process
with the classical Expectation-Maximization algorithm.

The document is very well written and pedagogical.

**Weaknesses:**

The experiments are rather disappointing. The two synthetic datasets
are toy examples (only two components mixtures). More over, contrary
to the affirmation in the introduction, only synthetic experiments are
shown and no real datasets are presented.

Moreover, this omission is highly problematic: on such an important
matter, whether it is a LLM hallucination or an oversight, it shows a
lack of seriousness and respect for the reader. The authors are visibly
unable to seriously proofread their article for this kind of mistakes...

Misc remarks:
- Please increase the size of figures
- Put Table 1&2 in the body, examples should not be second-class
  content

**Questions:**

What is the point of the paragraph "Expert knowledge" ? There is not
mention of it anywhere.

Where are the real datasets experiments ? You mention it in the
abstract, in the introduction, and at the beginning of the
experimental section but there are no results displayed.

Bayesian Optimization is known to be subject to the cold-start
problem. Are you affected and how do you deal with this issue ?

---

### Meta-Review · Area_Chair_scQe · 2026-01-13

**Summary:**

This submission studies mixtures of elliptical distributions and investigates the effectiveness of Bayesian optimization (BO) for estimation in these models. The main results show (a) global convergence to the MLE and (b) by BO achieves the optimal misclassification rate.

One reviewer expressed concerns about misrepresenting the actual contributions, which appear to be corroborated but could simply have been an oversight.

**Reviewer Concerns:**

Authors did not respond.

**Reviewer Scores:**

No changes.

---

### Decision · Program_Chairs · 2026-01-26

Reject